# Automatic detection of expressed emotion from Five-Minute Speech Samples: Challenges and opportunities

**Bahman Mirheidari** [1]☯*, **André Bittar** [2]☯*, **Nicholas Cummins** [2,3], **Johnny Downs** [3], **Helen L. Fisher** [4,5], **Heidi Christensen** [1]

**1** Department of Computer Science, University of Sheffield, Sheffield, United Kingdom, **2** Department of Biostatistics & Health Informatics, Institute of Psychiatry, Psychology & Neuroscience, King's College London, London, United Kingdom, **3** CAMHS Digital Lab, Department of Child and Adolescent Psychiatry, Institute of Psychiatry, Psychology & Neuroscience, King's College London, London, United Kingdom, **4** Social, Genetic & Developmental Psychiatry Centre, Institute of Psychiatry, Psychology & Neuroscience, King's College London, London, United Kingdom, **5** ESRC Centre for Society and Mental Health, King's College London, London, United Kingdom

☯ These authors contributed equally to this work.
* b.mirheidari@sheffield.ac.uk (BM); andre.bittar@kcl.ac.uk (AB)

**Data Availability Statement:** Due to the potentially identifying nature of these data, they are not publicly available. This and other E-Risk data can be accessed for free by researchers through a

## Abstract

Research into clinical applications of speech-based emotion recognition (SER) technologies has been steadily increasing over the past few years. One such potential application is the automatic recognition of expressed emotion (EE) components within family environments. The identification of EE is highly important as they have been linked with a range of adverse life events. Manual coding of these events requires time-consuming specialist training, amplifying the need for automated approaches. Herein we describe an automated machine learning approach for determining the *degree of warmth*, a key component of EE, from acoustic and text natural language features. Our dataset of 52 recorded interviews is taken from recordings, collected over 20 years ago, from a nationally representative birth cohort of British twin children, and was manually coded for EE by two researchers (inter-rater reliability 0.84–0.90). We demonstrate that the degree of warmth can be predicted with an $F_1$-score of **64.7%** despite working with audio recordings of highly variable quality. Our highly promising results suggest that machine learning may be able to assist in the coding of EE in the near future.

## 1 Introduction

*Expressed emotion* (EE) within the family environment has been extensively studied through asking caregivers to speak freely about a relative or family member in their care [1, 2]. More specifically, EE refers to the attitudes of a caregiver towards their child which can comprise both negative emotions, such as hostility, criticism, and/or emotional over-involvement, as well as positive emotions, such as warmth. Levels of EE within families have been studied for over five decades and have been shown to be a powerful transdiagnostic predictor of adverse

managed access process requiring an E-Risk Study sponsor. Full information on how to apply for access is available here: https://eriskstudy.com/data-access/ and queries should be emailed to Professor Helen Fisher at this address: eriskstudy@kcl.ac.uk.

**Funding:** This project was funded by the Psychiatry Research Trust (https://www.psychiatryresearchtrust.co.uk/) [39C] and UK MRC (https://www.ukri.org/councils/mrc/) [MR/X002721/1]. NC is part funded by the National Institute for Health Research (NIHR, https://www.nihr.ac.uk) Maudsley Biomedical Research Centre at South London and Maudsley NHS Foundation Trust and King's College London. JD received support from a National Institute of Health Research (NIHR) Clinician Scientist Fellowship [CS-2018-18-ST2-014] and Psychiatry Research Trust Peggy Pollak Research Fellowship in Developmental Psychiatry. HLF is part supported by the Economic and Social Research Council (ESRC, https://www.ukri.org/councils/esrc) Centre for Society and Mental Health at King's College London [ES/S012567/1]. The funders had no role in study design, data collection and analysis, decision to publish, or preparation of the manuscript.

**Competing interests:** The authors have declared that no competing interests exist.

outcomes across the lifecourse [3–5], for example predicting recovery and relapse patterns for adolescents with mood, anxiety, and eating disorders [6, 7], or adults with severe mental disorders [8]. Despite being a well-established clinical construct, EE has received very little attention from the affective computing community, whose primary focus is utilising artificial intelligence for the task of emotion recognition [9]. This is surprising, as the tools and automated methodologies could go some way to relieving the labour and training burden of using humans to code expressed emotion [10]. Affective computing applications are continually being recognised as being useful in clinical settings; e. g. [11, 12]. However, very few actual speech-based approaches can be found in the literature; prominent examples include the inference of *Attachment Condition* [13], assessing emotional engagement in dementia patients [14], and assisting in autism therapies [15].

There is a rich set of studies focusing on the task of general emotion detection from speech; e. g. [16–19]. *Speech emotion recognition* SER can be be interpreted as a regression problem or a classification model. As a regression model, the aim is to predict the emotion primitives such as arousal, valence, and dominance. For classification, they directly predict discrete emotion categories such as anger, happiness or sadness. A wide array of different machine learning modelling approaches can be seen in the SER literature. Many SER works, especially those undertaken on small databases, are still based on conventional classifiers; e.g. Support Vector Machines (SVM) and k-nearest neighbours (KNN)[18]. There are also a growing number of works utilising deep learning approaches, including convolutional neural networks (CNNs) and Transformers; however, database size and overfitting concerns in many SER applications [17, 18].

We have previously demonstrated that SER analytical pipelines can be adapted to aid the detection of EE constructs [20]. EE was originally measured through in-depth face-to-face interviews but, due to time constraints, has subsequently been assessed through brief samples of caregivers speaking freely about their child. These interactions are known as *Five-Minute Speech Samples* (FMSSs) [21]. Coding of EE focuses on the emotions that are apparent when the caregiver speaks about their child, drawing both on the content of what is said and the tone of voice that is used. Importantly, this coding contains clinically important information; EE rated from maternal speech samples has been shown to be associated with the development of antisocial behavioural problems in children [22] and subsequent serious mental illnesses [23]. Negative emotions from parents' speech have been demonstrated to be predictive of the onset and course of other mental health problems in children, including anxiety, depression, and attention-deficit hyperactivity disorder [24] underlining its usefulness as an early predictor of youth mental health difficulties.

The coding of EE is, however, labour-intensive and requires highly trained raters [10]. Even after training, human ratings potentially have limited reproducibility as they can be prone to drift and unconscious biases. The work presented in this paper is an extension of our preliminary analysis presented in [20]. As in that analysis, this current analysis is based on an automated approach for determining the degree of warmth, a key component of EE, from acoustic and text features. We build on our previous efforts with an increased sample size, and present a wider set of analyses. Automating the assessment of EE could dramatically impact clinical practice, by providing clinicians with an important indication as to the likelihood that a young person will develop mental health problems and enable them to effectively target preventive interventions and reduce incidence rates of mental disorders.

The overall aim of this paper is to assess the efficacy of standard affective computing processing pipelines typically used for speech-based emotion recognition (SER) [19] to classify *level of warmth* in our audio files. We chose warmth as our initial EE construct to investigate by drawing an analogy to arousal, which is arguably more straightforward to detect using voice

samples [25]. Warmth is assessed by EE raters according to both sentiment and tone [22], allowing us to test linguistic and acoustic speech markers, as well as a combined solution. In particular, we assess acoustic features extracted through the OPENSMILE Toolkit [26]; and text features (Term Frequency and Word Embeddings) extracted from manual and automatic (utilising automatic speech recognition (ASR)) transcriptions of our audio files. As warmth is expressed in both content and acoustics, we explore combinations of these modalities as well. Given the small size of our data files, we classify these features using four conventional machine learning techniques: Logistic Regression, Support Vector Machine, Random Forest and a Multi-Layer Perceptron.

## 2 Materials and methods

### 2.1 Cohort study

The Environmental Risk (E-Risk) Longitudinal Twin Study tracks the development of a nationally representative birth cohort of 2,232 British twin children born in England and Wales in 1994-1995. Briefly, the E-Risk sample was constructed in 1999-2000, when 1,116 families (93% of those eligible) with same-sex 5-year-old twins participated in home-visit assessments. This sample comprised 56% monozygotic (MZ) and 44% dizygotic (DZ) twin pairs; sex was evenly distributed within zygosity (49% male). The study sample represents the full range of socioeconomic conditions in Great Britain, as reflected in the families' distribution on a neighbourhood-level socioeconomic index A Classification of Residential Neighbourhoods (ACORN) developed by Consolidated Analysis Center, Inc. (CACI) for commercial use. E-Risk families' ACORN distribution closely matches that of households nation-wide: 25.6% of E-Risk families live in "wealthy achiever" neighbourhoods compared to 25.3% of households nation-wide; 5.3% vs 11.6% live in "urban prosperity" neighbourhoods; 29.6% vs 26.9% live in "comfortably off" neighbourhoods; 13.4% vs 13.9% live in "moderate means" neighbourhoods; and 26.1% vs 20.7% live in "hard-pressed" neighbourhoods. E-Risk under-represents urban prosperity neighbourhoods because such households are likely to be childless.

The twins have been comprehensively assessed during home visits at ages 5, 7, 10, 12 and 18 years (with 93% retention). The Joint South London and Maudsley and Institute of Psychiatry Research Ethics Committee approved each phase of the study. Parents gave written informed consent, and twin participants gave written assent at ages 5–12 and written informed consent at age 18. Further details are reported elsewhere [27].

### 2.2 Assessment of EE

When the children were 5 years old, speech samples of approximately five minutes were audio-recorded from caregivers (almost exclusively mothers) in their homes to elicit expressed emotion about each child. Trained interviewers asked caregivers to describe each of their children ("For the next 5 minutes, I would like you to describe [child] to me; what is [child] like?"). The caregiver was encouraged to talk freely but if s/he found this difficult, a series of semi-structured probes were used (e.g., "In what ways would you like [child] to be different?"). Interviews about each twin were separated in time by approximately 90 minutes. All interviews were audio-taped with the caregiver's consent. Data for expressed emotion were missing for 9% of the sample because some caregivers did not wish to be audio-taped or because of technical problems with the tape.

These speech samples were coded by two trained raters according to guidelines adapted from the FMSS scoring manual and modified for use with preschool children [22]. The raters underwent 2 weeks of training about coding expressed emotion. Inter-rater reliability was established by having the raters individually code audiotapes describing 40 children. High

inter-rater reliability (r = 0.84–0.90) was established [22]. The same rater coded both twins in the same family. The rater was blind to all other E-Risk Study data. Ratings of EE included the degree of warmth that the caregiver expressed towards each child.

Warmth was a global measure used to describe the whole speech sample. The scale refers only to the warmth expressed in the interview about the child. The warmth of the respondent's personality was not a consideration, nor was warmth shown toward others. Positive comments in themselves were not viewed as evidence of warmth, nor were stereotyped endearments. Warmth was assessed by the tone of voice, spontaneity (e.g., "She is so funny—the other day she made up a song and she was dancing and singing in the garden . . . the song was about everything . . . a butterfly flew by and that ended up in the song . . . it was so sweet."), sympathy and/or empathy toward the child (e.g., "I feel really sorry for her, it is not her fault . . . I worry for her."). Warmth was coded on a 6-point scale. High warmth (5) and moderately high warmth (4) were coded when there was definite and clear-cut tonal warmth, enthusiasm, interest in, and enjoyment of the child. For example, "She is a delight, she is so happy, I love taking her out, she is my ray of sunshine" was coded as a 5. Moderate warmth (3) was coded when there was definite understanding, sympathy, and concern but only limited warmth of tone, for example, "I worried about her when she went to school, I thought she may have difficulty in mixing, and I felt sorry for her." Some warmth (2) was coded when the mother showed a detached, rather clinical approach and little or no warmth of tone, but moderate understanding, sympathy, and concern. For example, an interview with comments along the lines of "She's alright" with little substantiation would have received this rating. Very little warmth (1) was rated when there was only a slight amount of understanding, sympathy, concern, enthusiasm about, or interest in the child. No warmth (0) was reserved for mothers who showed a complete absence of the qualities of warmth as defined. The inter-rater agreement for warmth was .90 [22].

## 2.3 Speech data

The interviews from the E-Risk study were recorded on cassette tapes and stored over 20 years. Over time the quality of the recordings degraded and they contain a significant amount of inaudible passages. Recently we have used professional equipment to digitise the tapes.

The interviewers followed semi-structured conversation protocols and hence the conversations compromise overlapping speech and background noises. To train an automatic speech recognition (ASR), the interviews needed to be transcribed, however, due to budget concerns, a limited number of the interviews were transcribed.

Due to the audio quality issues, the transcription still contained inaccurate segmentation and missing sections with numerous incorrect words. Finally, 52 out of around a thousand recordings (104 regarding twin children) were coded by the raters.

## 2.4 Analysis

Since we focus on classifying the level of warmth in the interviews, we focus on validating the classification efficiency using different types of features, including acoustic-only, text-only and combinations of both.

To align the audio segments to the speakers, Audacity (https://www.audacityteam.org) was used. We used different tags to assign the segments of the interviewers and the mothers to *general talk* about both twins (e.g., the level of support during pregnancy), and *specific talk* about the elder and younger twins (e.g., feeling about her elder twin). In total, we had 38 distinct tags (19 for interviewers and 19 for mothers).

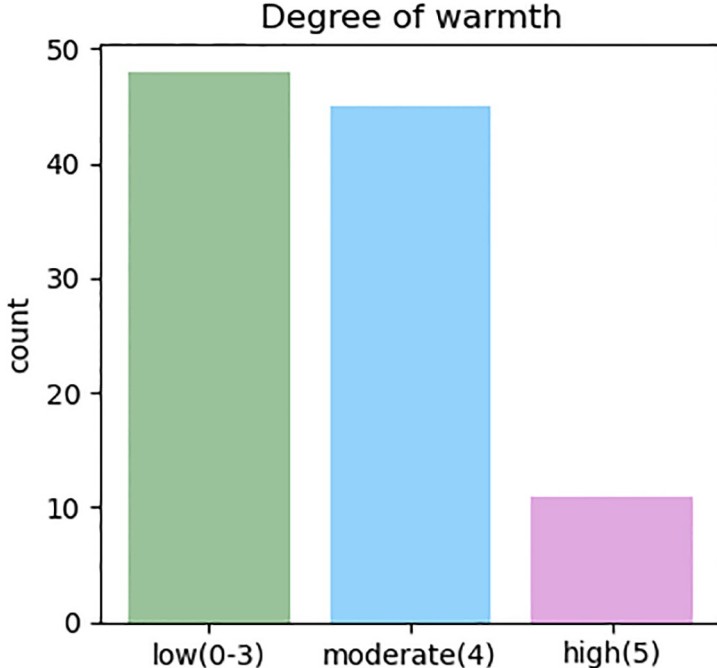

**Fig 1. Distribution of caregiver warmth classes in our 3-way schema with corresponding 6-way schema numerical classes.**

Using the tags for the elder and younger twins, we divided the 52 recordings into 104 samples.

Inaudible segments of the speech data were ignored. Due to having strong environmental background noises, we could not apply any noise reduction technique (causing loss of acoustic information).

The coding of warmth includes 6 ordinal classes (0 to 5). However, the distribution was imbalanced across the 104 samples. Thus, we merged the classes into 3 classes to obtain a more balanced distribution. The final distribution of the code classes is shown in Fig 1.

## 3 Results

Four different classical machine learning classification models were used (provided by [28] Python library) to be trained on the data and used to predict the level of warmth: *Logistic Regression* (LR), *Linear Support Vector Classifier* (Lin-SVC), *Random Forest* (RF), and *Multi-Layer Perceptron* (MLP). Evaluation of the models was carried out by running five times 10-fold shuffled cross-validation. The evaluation metric was $F_1$-score of the classifiers.

### 3.1 Acoustic features

The OPENSMILE Toolkit [26] was used to extract a few frame-based acoustic features from the mother's audio segments. We then took the average, standard deviation, minimum, maximum and sum of the features and made fixed-length feature sets for training the classifiers. The features used were previously introduced at various Interspeech challenges on emotion detection including the *Interspeech 2009 Emotion Detection Challenge* (IS09-EMO) [29], the *Audio-Visual Emotion recognition Challenge 2013* (AVEC13) [30], Interspeech 2010, 2013 and 2016

**Table 1. Average $F_1$-score and standard deviation (10 runs, 10-fold cross validation) of the four classifiers using different acoustic-only features.**

| Features | LR(%) | Lin-SVC(%) | RF(%) | MLP (%) |
|:---:|:---:|:---:|:---:|:---:|
| AVEC13 | 35.6(2.6) | 39.9(3.3) | **64.1(2.8)** | 39.1(4.9) |
| IS09-EMO | 44.8(2.8) | 47.2(3.4) | **63.2(2.7)** | 38.9(4.3) |
| IS10-CPE | 46.9(3.6) | 45.5(3.2) | **63.9(2.1)** | 42.5(4.9) |
| IS13-CPE | 36.9(2.6) | 40.9(3.5) | **64.3(2.5)** * | 39.5(4.6) |
| IS16-CPE | 37.2(2.9) | 40.9(3.5) | **63.7(3.2)** | 40.7(5.9) |
| eGeMAPS | 42.7(3.6) | 39.6(4.1) | **64.0(2.5)** | 37.0(3.7) |
| AVG(STD) | 40.7(4.3) | 42.3(2.9) | **63.9(0.4)** | 39.6(1.7) |

The best result in a row is in bold.

\*:The best overall result.

*Computational Paralinguistics ChallengE* (ComParE) (IS10-CPE, IS13-CPE, IS16-CPE) [31–33], and the *extended Geneva Minimal Acoustic Parameter Set* (eGeMAPS) [34]. These feature representations are omnipresent throughout the SER literature to the point they represent a pseudo-standard [16–19].

Table 1 shows the average $F_1$-score and standard deviation (SD) of the four classifiers on different acoustic features. The results varied across combinations of features and classifiers. The RF classifier using IS13-CPE features achieved the best overall $F_1$-score of 64.3% (SD:2.5%), while, the LR classifier using AVEC13 achieved the worst overall $F_1$-score of 35.6% (SD:2.6%). However, the RF classifier achieved the highest average $F_1$-score of 63.9% (SD:0.4%) for all individual acoustic features (last row). In brief, the results of the other three classifiers were significantly worse than the RF classifier (e.g. the average $F_1$-score on all features for LIN-SVC was 21.6% less than for the RF).

Fig 2 shows the Confusion Matrix (CM) of a representative (chosen as the classifier with the $F_1$-score close to the overall average in 10 runs). RF classifier using IS13-CPE features. Over 67% (32 out of 48) of the 'warmth low', 64% (7 out of 11) of the 'warmth high' and 62% (28 out of 45) of the 'warmth moderate' were classified correctly. One-third of the 'warmth moderate' scores, however, were confused with the 'warmth low', and 31% of the 'warmth low' scores were misclassified as the 'warmth moderate'.

## 3.2 Text features from manual transcriptions

A number of different textual features were extracted from the transcriptions including the Scikit-Learn `TfIdfVectorizer` [28], pre-trained word embeddings: GloVe [35] 25 dimensions, FastText [36] 300 dimensions, trained on Wikipedia (FstTxt), Word2Vec [37] 300 dimensions, trained on the Google News corpus (W2Vec), as well as pre-trained transformer-based language models (large uncased BERT [38], and large uncased RoBERTa). These feature representations were chosen as they are widely used in both SER and text-based sentiment tasks; e. g. [39–42].

The spaCy [43] toolkit is used for tokenising the text, removing punctuation and lingering whitespace, and lowercasing all tokens. For pre-trained language models, the length of the sequence is limited to 512 tokens with 50% overlap and the average and standard deviation of the last three layers was computed to create features for classification. For the TF-IDF and pre-trained language models, we trialled using the original transcriptions with punctuation (reported in results with the suffix "*-pun*") and the pre-processed texts. The unknown words

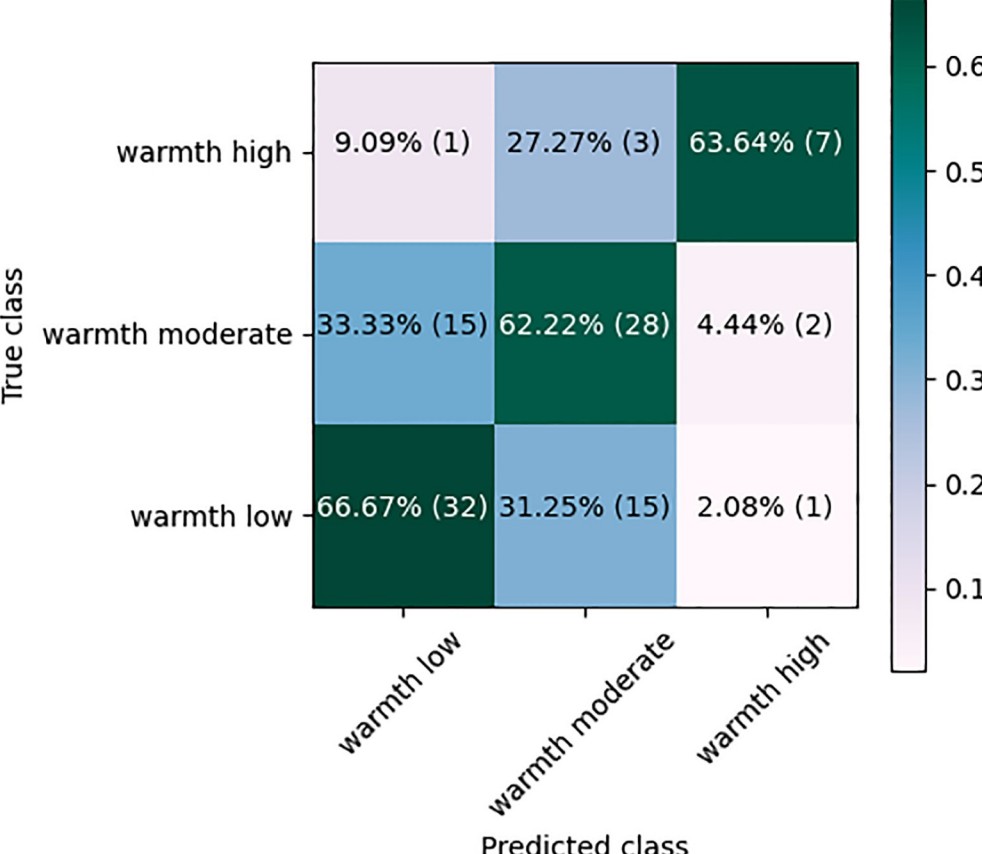

**Fig 2. Confusion matrix of the RF classification using IS13-CPE acoustic features.**

(words that did not appear in the model) were ignored in word embedding and the final embedding was the mean of embeddings for all known word tokens in the transcript. We used the same classification models as for the acoustic features.

Table 2 shows the average $F_1$-score and SD of the four classifiers trained on different text features. The best average $F_1$-score was achieved by a Lin-SVC classifier using the BERT features (60.4%, SD: 2.7%), while the lowest performance was obtained with the LR classifier using FstTxt features ($F_1$-score of 39.5%, SD: :2.8%). The Lin-SVC classifier obtained the best results on all individual text features (average $F_1$-score of 52.5%, last row) followed by the LR classifier (average $F_1$-score of 51.6%). In brief, the results on individual text features were not significantly different among the four classifiers, in contrast to the acoustic features (Table 1), where the RF classifier's results were significantly better than the others. Also, the results on text features with punctuation were not significantly better than the corresponding features without punctuation (e.g. for the Lin-SVC classier, BERT: 60.4% versus BERT-PUN: 59.5%; RoBERTa: 55.8% versus RoBERTa-PUN: 57.8%). The best classifier using the BERT features (Lin-SVC) achieved around 4% lower average $F_1$-score compared to the best classifier using the IS13-CPE features (RF), i.e. 60.4% versus 64.3%. This caused much more misclassifications between the three classes. Fig 3 shows the corresponding CM with more confusions, e.g. 78% of the 'warmth moderate' were confused with the 'warmth low' scores, and only 55% (6 out of 11) of the 'warmth high' were classified correctly.

**Table 2. Average $F_1$-score and standard deviation (10 runs, 10-fold cross validation) of the four classifiers using different text-only features.**

| Features | LR(%) | Lin-SVC(%) | RF(%) | MLP (%) |
|---|---|---|---|---|
| TF-IDF | 52.1(2.3) | 51.2(2.7) | **53.5(3.1)** | 49.6(2.9) |
| TF-IDF-PUN | 52.0(2.3) | 51.4(3.2) | **52.9(3.3)** | 49.5(2.9) |
| BERT | 58.9(2.5) | **60.4(2.7)** * | 50.1(3.4) | 53.9(3.3) |
| BERT-PUN | 57.0(3.1) | **59.5(2.5)** | 49.8(1.1) | 54.4(1.7) |
| RoBERTa | **57.6(2.9)** | 55.8(2.0) | 52.0(2.2) | 52.6(2.4) |
| RoBERTa-PUN | 57.7(2.7) | **57.8(2.6)** | 48.8(3.1) | 55.6(3.3) |
| W2Vec | 46.5(1.9) | 47.3(3.1) | **48.7(2.7)** | 47.4(3.3) |
| FstTxt | 39.5(2.8) | 43.4(3.2) | 45.1(3.0) | **45.9(1.9)** |
| GloVe | 43.0(3.3) | 45.7(2.5) | **51.0(2.9)** | 46.2(2.6) |
| AVG(STD) | 51.6(6.7) | **52.5(5.9)** | 50.2(2.4) | 50.6(3.5) |

The best result in a row is in bold.

*:The best overall result.

## 3.3 Text features from automatic speech recognition

The text features in the previous section were extracted from the manual transcription of the recordings. In order to have a fully automatic system, in this section, we replace the manual text with automatic text produced by our trained automatic speech recognition (ASR) system. Due to having a limited number of audio recordings we applied the 5-fold cross validating approach, i.e. the audio recordings data was split into five folds and we trained five ASRs (4 folds for training and one for test). We used the LibriSpeech dataset (with over 1000 hours of speech in reading mode) to train a base Time Domain neural Networks acoustic model, following the Kaldi's LibriSpeech recipe [44]. Due to having a small amount of data, the AMI dataset [45] (recordings of the conversations in different meeting rooms) was then added to the training set of our dataset in each fold to boost the acoustic and language models of the ASRs. The "transferring all layers" technique [46] was applied to adapt the acoustic model of the LibriSpeech dataset to the training dataset (with conversational speech). For the language model, the four-gram model was used with Turing smoothing interpolated with the language model of the LibriSpeech text (50% weight for each language model). The average (Word Error Rate) WER achieved on all speakers' utterances was **57.4%**. The high error rate of the ASR reflects the challenging nature of the data (poor quality of the digitised tape recordings, high volume of background noise in the real crowdy hospital environment, and difficulties of conversational speech). The quality of significant amount of recordings were so poor that the transcribers also could not recognise some words.

We used the outputs of the ASRs to extract the same text features except for those with punctuation (our ASRs do not output punctuation). (TF-IDF-PUN, BERT-PUN and RoBERTa-PUN). Then similarly we ran the four classifiers 10 times and calculated the average $F_1$-scores. Fig 4 shows the average $F_1$-scores obtained using the text features extracted from the manual transcriptions compared to the text features extracted from the ASR outputs. As can be seen, using the erroneous ASR outputs decreased all the results dramatically. The figures were significantly lower for the TF-IDF, BERT and RoBERTa features across the four classifiers (e.g. the average $F_1$-score of the LIN-SVC classifier using the BERT features decreased around 10%, from 60.4% to 50.1%). Slightly lower decreases were observed for the W2Vec, FstTxt and GloVe features (e.g. the average $F_1$-score of the LIN-SVC classifier using the GloVe

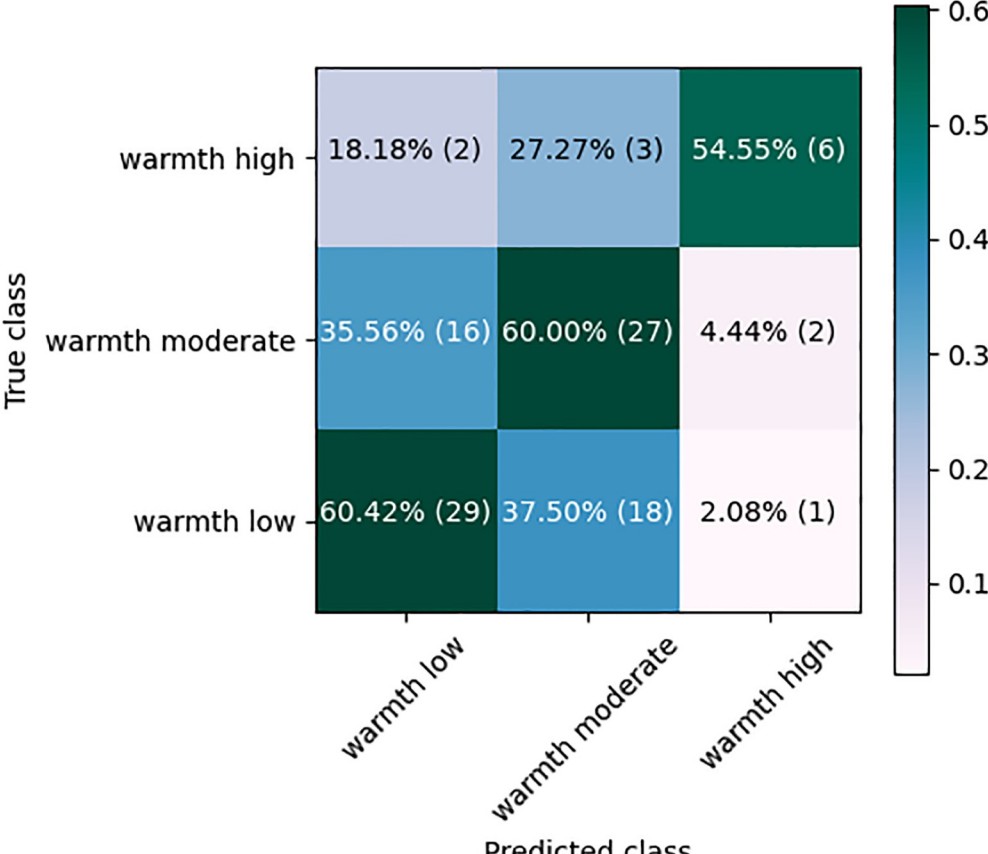

**Fig 3. Confusion matrix of the Lin-SVC classification using BERT features.**

feature decreased 1.7%, from 45.7% to 44%). For the LR, Lin-SVC and MLP classifiers, the results of the ASR-produced features were lower than features from the manual transcriptions, yet most of the text features yielded comparable or better results than the acoustic only features. The green dashed-dotted line shows the average $F_1$-score of using the IS13-CPE features and some of the ASR-produced results are above the line (e.g. MLP classifier using W2Vec from the ASR resulted in 44.7% $F_1$-score, 5.2% more than 39.5%, the line). However, for the RF classifiers, all the text features, including those extracted from the manual transcriptions had significantly lower $F_1$-scores, i.e. all below the green dotted line.

## 3.4 Combining acoustic and text features

The manual coding of warmth (and EE in general) relies on both interview content and voice features. We therefore sought to assess the use of both modalities in the classification task, using a combination of acoustic and text features to train the models. Since the IS13-CPE acoustic features yielded the best average results across all classifiers, we combined these features with each of the text features. The average $F_1$-score of the classifiers on the combined features using the manual transcriptions versus the outputs of the ASR are shown in Fig 5. Note that we normalised the features before combining them, since they had different ranges of values.

For the LR, Lin-SVC and MLP classifiers, combining the acoustic and text features, in most cases, resulted in better average $F_1$-scores compared to the acoustic-only and text-only

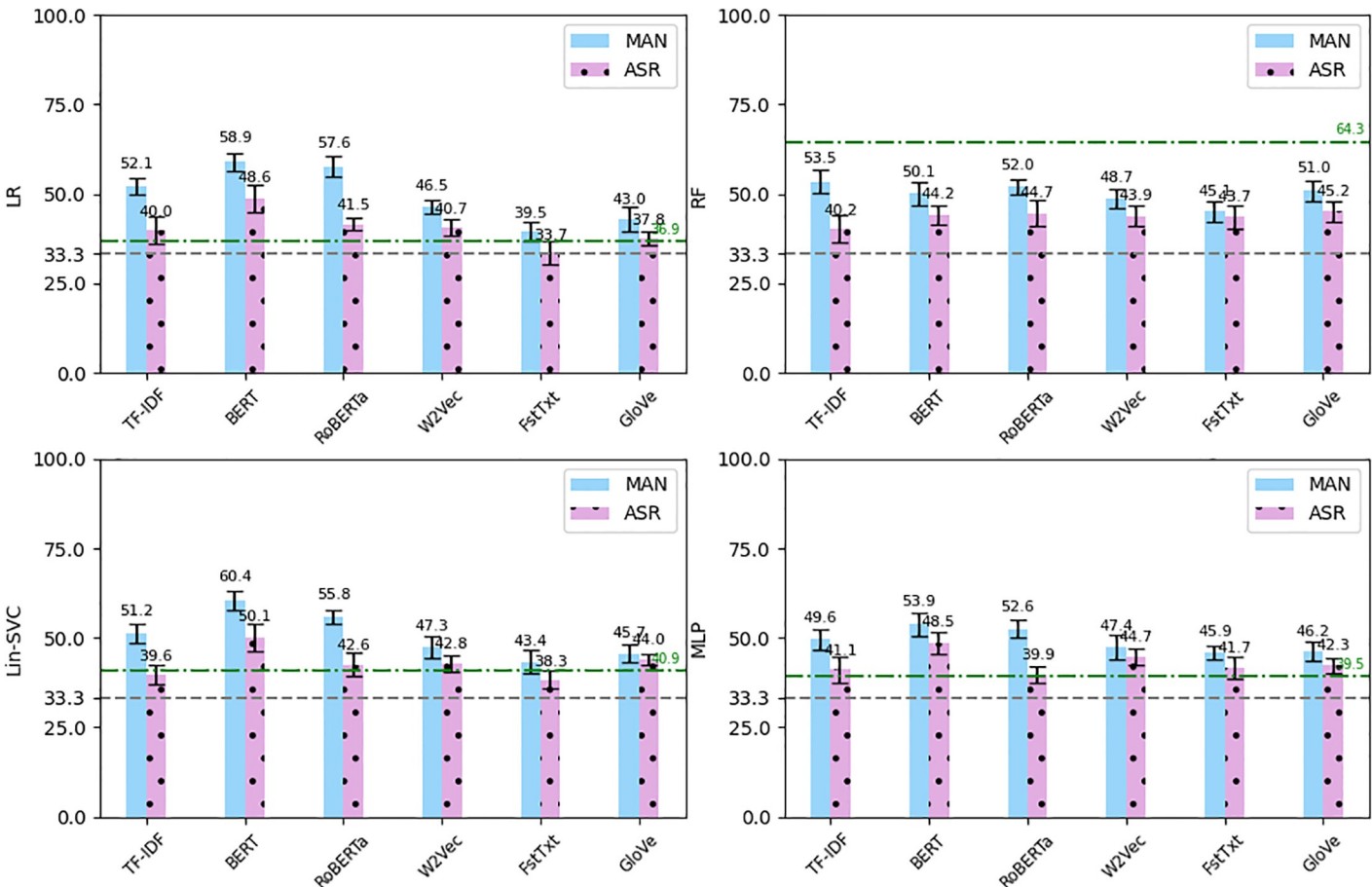

**Fig 4. The average $F_1$-score (error bars: Standard deviation) of the four classifiers using text features from manual transcriptions (MAN) versus the features from the ASR outputs.** Gray dashed line: Chance level, Green dashed-dotted line: $F_1$-score using IS13-CPE features.

features, though greater improvements were observed among the ASR-produced features. A few of them gained slightly better results than the corresponding text features from the manual transcriptions, e.g., the MLP classifier using the combined IS13-CPE features with GloVe from ASR achieved an $F_1$-score of 63.5%. However, for the RF classifier, combining features could not increase the $F_1$-score as using the acoustic-only feature (only IS13+Glove on the manual transcript and the ASR text gained close results between 64-64.5%). Since the RF classifier using the acoustic-only features had already the best classification results, adding the text features could not yield a better outcome.

Focusing on the RF classifier, in particular, we have tried two other fusion approaches: voting and using Principal Component Analysis (PCA) (since the text features like BERT and RoBERTa have high dimensions compared to the acoustic features, they might swamp the classifier) to reduce the dimensionality of the features. In voting, we trained three different classifiers a) on the acoustic-only features, b) on text-only features, and c) on the combined features. Then we applied voting among the three results obtained from the three classifiers, i.e., we calculated the mode of the three results. In addition, we applied PCA to the acoustic-only and text-only features to reduce the dimensionality of the feature sets. We used 50 components for PCA (for the GloVe features with 25 dimensions we did not apply PCA). We also tried only

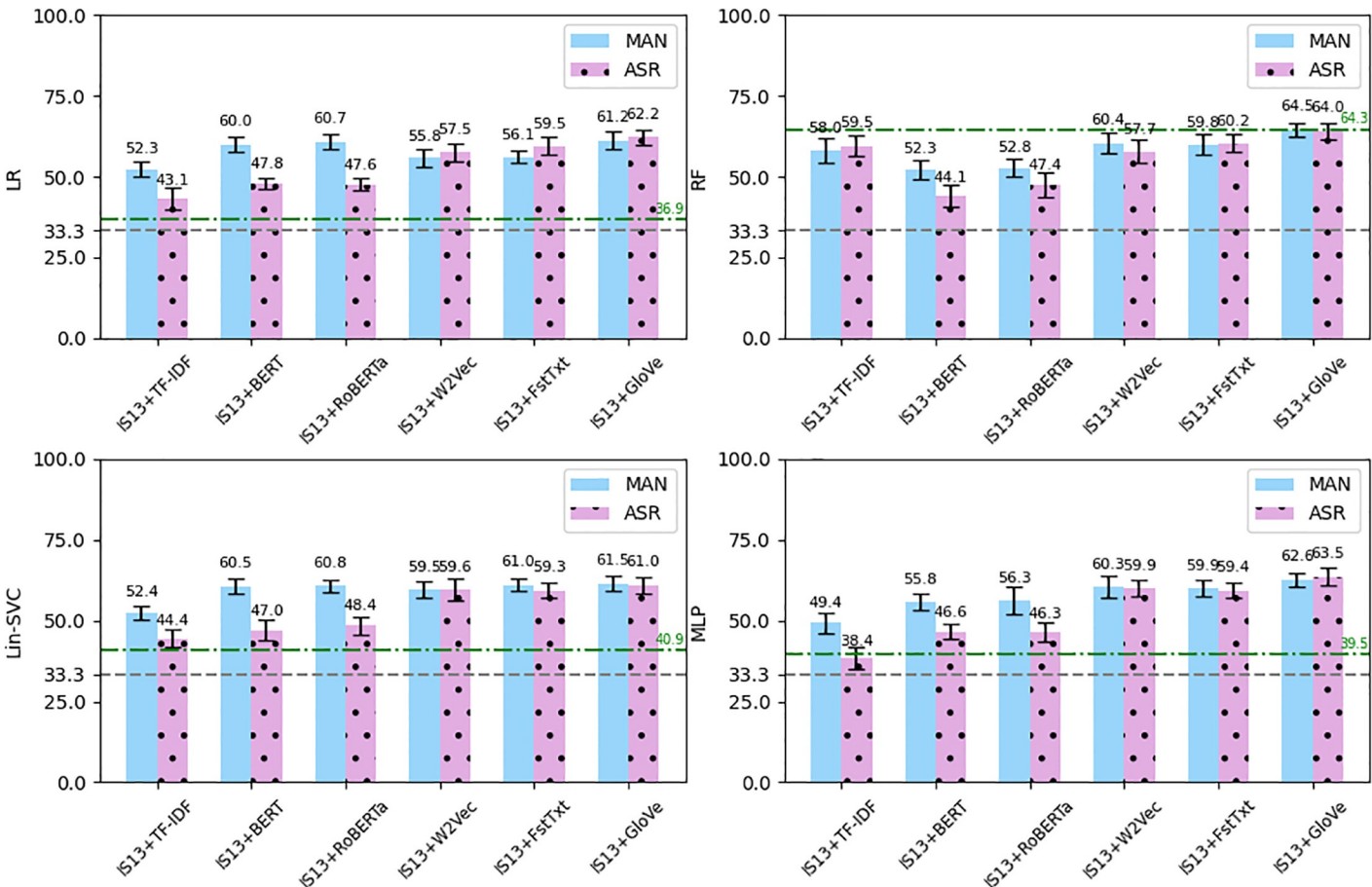

**Fig 5. The average $F_1$-score (error bars: Standard deviation) of the four classifiers using text features from manual transcriptions (MAN) combined with IS13-CPE features versus the features from the ASR outputs combined with IS13-CPE.** Gray dashed line: Chance level, Green dashed-dotted line: $F_1$-score using only IS13-CPE features.

PCA without voting but the results were not much better than having both. Fig 6 shows the results of the two approaches for the RF classifier. PCA and voting on the IS13+Glove features achieved $F_1$-score of 64.9% and 64.7% on manual text and ASR text respectively (0.6%, 0.4% improvements compared to 64.3% on using the acoustic-only features). So the approaches could only very slightly improve the overall result. Comparing to Fig 5, the approaches could improve the results on the other individual combined features, although they are still below the green dashed-dotted line representing the ($F_1$-score using only IS13-CPE features).

Fig 7 shows the CM of the RF classifier using the combined IS13+Glove features. Compared to Fig 2, the number of the 'warmth high' scores that are classified correctly increased from 7 to 8 (or from 64% to 73%). Also, some of the confusion between the classes changed.

## 4 Discussion

Our results highlight that combining acoustic and natural language feature representations with machine learning it is possible to achieve an $F_1$-score of **64.7%** when classifying the *degree of warmth*, a key component of expressed emotion in five-minute speech samples. To the best of the authors' knowledge, there is no similar speech emotion recognition research in the

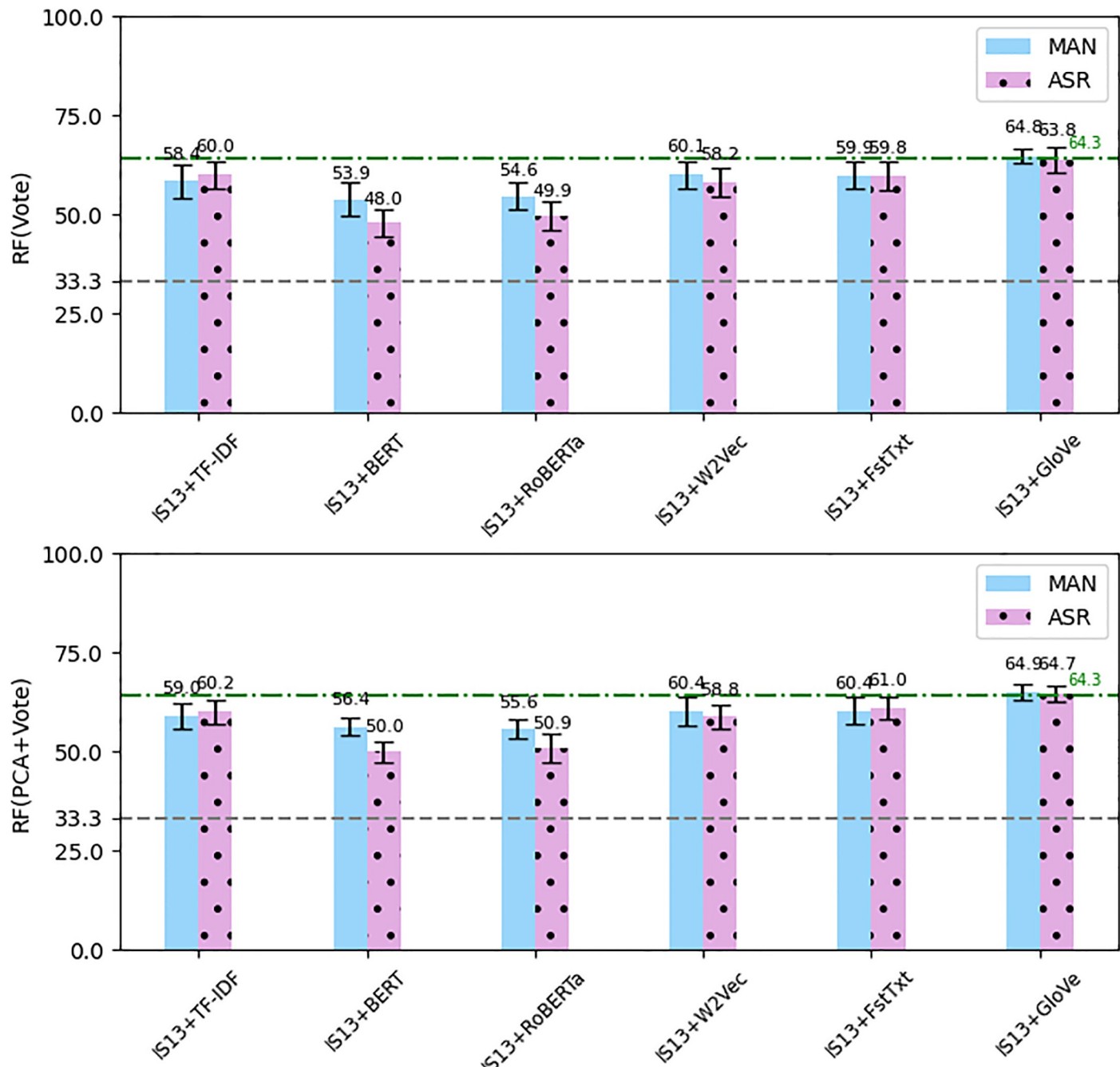

**Fig 6. The average $F_1$-score (error bars: Standard deviation) of RF classifier using combined text and IS13-CPE acoustic features from manual transcriptions (MAN) and the ASR outputs (ASR) applying voting approach (Vote) compared to applying PCA plus voting.** Gray dashed line: Chance level, Green dashed-dotted line: $F_1$-score using only IS13-CPE features.

literature (other than our preliminary results [20]) from which to draw meaningful comparisons. However, we believe, that given the challenging nature of our dataset in terms of audio quality, this is a highly promising result. These results offer us a strong baseline from which to build as we grow the size of our data. Additionally, they highlight the potential of using

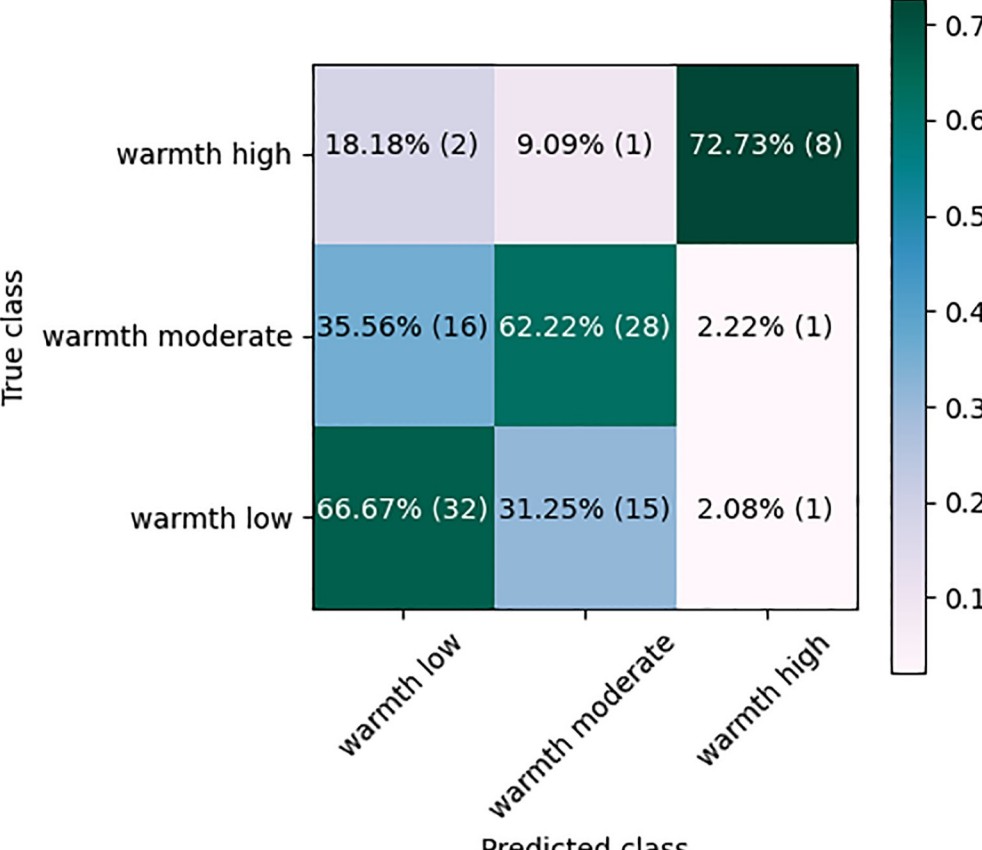

**Fig 7. Confusion matrix of the RF classification using combined IS13-CPE and Glove features from the ASR.**

machine learning classifiers to eventually substitute for the process of manual coding of warmth, and EE more generally, by human raters. Interestingly, despite the lower quality of our audio, our results demonstrate the strength of using acoustic features for this task; our strongest acoustic-only $F_1$-score was 64.3%, outperforming the strongest text-only $F_1$-score which was 60.4%. This observation is potentially a function of both data quality and size, meaning we could extract and model more meaningful acoustic representations than those obtained through our NLP pipelines. Future work will focus on improving the robustness and generalisability of our findings through the addition of more data and more complex modelling pipelines.

We have identified two main limitations in the presented work. First, we had significant challenges to overcome associated with the real-world nature of our dataset. This included our data being originally recorded on low-quality (by today's standard) cassette tapes, with low-grade recording equipment. This impacted the quality and amount of transcription we had available to use, most likely impacting the performance of our NLP system. In [20], we make a set of recommendations for working with such data. These include investing time in the manual preparation of data and using low-complexity modelling. In this regard, our second limitation is that we did not utilise more contemporary deep learning models. This was a deliberate design choice given the size of our dataset meant it was simply unfeasible to trial state-of-the-art models that require large amounts of training data. Highlighting this, is that in our study

the RF classifier did not build better decision-making models with the addition of text features to the acoustic features. Due to the limited sample size, we are not confident that our results will hold as we expand the size of our database. Despite these limitations, our results tentatively indicate that combining acoustic and text features is optimal when trying to predict the levels of caregiver warmth expressed in Five Minute Speech Samples.

In future work, we intend to prioritise the expansion of the dataset with additional transcriptions. A larger dataset would open up the possibility of using more sophisticated classification models. So far, we have extracted only a narrow set of acoustic and text features to train our classifiers, providing scope to explore additional acoustic, text and linguistic features in further work. Additionally, in this study, we tried only two fusion approaches: combining features (an early fusion strategy) and voting (a late fusion strategy); again leaving scope for exploring other fusion in future (such as attentive fusion). We also aim to develop an approach based on automatic speech recognition in order to alleviate the burden of manual transcription. We believe these additions will make considerable gains in the development of automated tools for expressed emotion annotations capable of working on a wide range of vary quality audio files.

## Acknowledgments

We are grateful to the E-Risk study families and twins for their participation. Our thanks to Professors Terrie Moffitt and Avshalom Caspi, the founders of the E-Risk study, Professor Louise Arseneault who led the phase 18 data collection, and to the E-Risk team for their dedication, hard work, and insights.

## Author Contributions

**Conceptualization:** Nicholas Cummins.

**Data curation:** Helen L. Fisher.

**Formal analysis:** Bahman Mirheidari, André Bittar.

**Software:** Bahman Mirheidari.

**Supervision:** Johnny Downs, Helen L. Fisher.

**Validation:** Nicholas Cummins, Johnny Downs, Helen L. Fisher, Heidi Christensen.

**Writing – original draft:** Bahman Mirheidari, André Bittar, Nicholas Cummins, Johnny Downs, Helen L. Fisher, Heidi Christensen.

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
