## [Decision Letter · Decision Letter 0]

5 Jan 2024

PONE-D-23-09462Automatic detection of expressed emotion from Five-Minute Speech Samples: Challenges and opportunitiesPLOS ONE

Dear Dr. Mirheidari,

Thank you for submitting your manuscript to PLOS ONE. After careful consideration, we feel that it has merit but does not fully meet PLOS ONE’s publication criteria as it currently stands. Therefore, we invite you to submit a revised version of the manuscript that addresses the points raised during the review process.

We look forward to receiving your revised manuscript.

Kind regards,

Jin Liu

Academic Editor

PLOS ONE

Journal Requirements:

“This project was funded by the Psychiatry Research Trust (https://www.psychiatryresearchtrust.co.uk/) [39C] and UK MRC (https://www.ukri.org/councils/mrc/) [MR/X002721/1]. NC is part funded by the National Institute for Health Research (NIHR, https://www.nihr.ac.uk) Maudsley Biomedical Research Centre at South London and Maudsley NHS Foundation Trust and King’s College London. JD received support from a National Institute of Health Research (NIHR) Clinician Scientist Fellowship [CS-2018-18-ST2-014] and Psychiatry Research Trust Peggy Pollak Research Fellowship in Developmental Psychiatry. HLF is part supported by the Economic and Social Research Council (ESRC, https://www.ukri.org/councils/esrc) Centre for Society and Mental Health at King's College London [ES/S012567/1].”

3. For studies involving third-party data, we encourage authors to share any data specific to their analyses that they can legally distribute. PLOS recognizes, however, that authors may be using third-party data they do not have the rights to share. When third-party data cannot be publicly shared, authors must provide all information necessary for interested researchers to apply to gain access to the data. (https://journals.plos.org/plosone/s/data-availability#loc-acceptable-data-access-restrictions)

a) A description of the data set and the third-party source

b) If applicable, verification of permission to use the data set

c) Confirmation of whether the authors received any special privileges in accessing the data that other researchers would not have

d) All necessary contact information others would need to apply to gain access to the data.

Additional Editor Comments (if provided):

This manuscript can be accepted for publication after minor edits:

1. more detail regarding how the data is preprosessed for the four machine learning algorithms should be presented in 2.4.

2. suggestions by the reviewers,

In the meantime, the authors should also make sure the dataset used for this research can satisfy the requirement of Plos One as stated at: https://journals.plos.org/plosone/s/data-availability

Reviewers' comments:

Reviewer's Responses to Questions

**Comments to the Author**

1. Is the manuscript technically sound, and do the data support the conclusions?

Reviewer #1: Yes

Reviewer #2: Partly

2. Has the statistical analysis been performed appropriately and rigorously? 

Reviewer #1: Yes

Reviewer #2: Yes

3. Have the authors made all data underlying the findings in their manuscript fully available?

Reviewer #1: Yes

Reviewer #2: No

4. Is the manuscript presented in an intelligible fashion and written in standard English?

Reviewer #1: Yes

Reviewer #2: Yes

5. Review Comments to the Author

Reviewer #1: This is a fine paper. The methodology is worked out in a good way. They do not overinterpret their findings. The conclusions follow from the data and analysis.

The sample is somewhat small and the data are a bit old, but these aspects are indicated in the discussion. Results are promising indeed.

Reviewer #2: more literature review and recent papers can be refered . Technicall the Automatic detection of expressed emotion from Five Minute Speech is good . The result based insights can be included in the conclusion. In the experiment, the manuscript compares the results against Existing work.

It needs to justify why specifically these proposed work is compared

6. PLOS authors have the option to publish the peer review history of their article (what does this mean?). If published, this will include your full peer review and any attached files.

Reviewer #1: No

Reviewer #2: No

---

## [Author Response · Author response to Decision Letter 0]

20 Feb 2024

We appreciate the editor and the reviewers' invaluable comments and suggestions. We have amended the manuscripts according to the comments. In the following paragraphs, the original comments are written in italics, and the author's corresponding answers are in bold.

Journal Requirements: 

Author response: Thanks for reminding us of this, we have already followed the latex template of the PLOS ONE and the suggested style (https://journals.plos.org/plosone/s/latex) and have named our files in line with the journal's style requirements.

“This project was funded by the Psychiatry Research Trust (https://www.psychiatryresearchtrust.co.uk/) [39C] and UK MRC (https://www.ukri.org/councils/mrc/) [MR/X002721/1]. NC is part funded by the National Institute for Health Research (NIHR, https://www.nihr.ac.uk) Maudsley Biomedical Research Centre at South London and Maudsley NHS Foundation Trust and King’s College London. JD received support from a National Institute of Health Research (NIHR) Clinician Scientist Fellowship [CS-2018-18-ST2-014] and Psychiatry Research Trust Peggy Pollak Research Fellowship in Developmental Psychiatry. HLF is part supported by the Economic and Social Research Council (ESRC, https://www.ukri.org/councils/esrc) Centre for Society and Mental Health at King's College London [ES/S012567/1].”

Author response: Thank you for the opportunity to amend our funding statement. The amended version which now fully conforms to PLOS One guidelines is as follows:

“This project was funded by awards from the Psychiatry Research Trust (https://www.psychiatryresearchtrust.co.uk/) [grant number: 39C] to JD and HLF and the UK Medical Research Council (https://www.ukri.org/councils/mrc/) [grant number: MR/X002721/1] to JD. The E-Risk Study is funded by the UK Medical Research Council (https://www.ukri.org/councils/mrc/) [grant numbers: G1002190 and MR/X010791/1] to HLF. NC is part funded by the National Institute for Health and Care Research (https://www.nihr.ac.uk) Maudsley Biomedical Research Centre at South London and Maudsley NHS Foundation Trust and King’s College London. JD received support from a National Institute of Health and Care Research (https://www.nihr.ac.uk) Clinician Scientist Fellowship [grant number: CS-2018-18-ST2-014] and Psychiatry Research Trust (https://www.psychiatryresearchtrust.co.uk/) Peggy Pollak Research Fellowship in Developmental Psychiatry. HLF is part supported by the Economic and Social Research Council (https://www.ukri.org/councils/esrc) Centre for Society and Mental Health at King's College London [grant number: ES/S012567/1]. The views expressed are those of the authors and not necessarily those of the UK Medical Research Council, Economic and Social Research Council, National Institute for Health and Care Research, the Department of Health and Social Care, the University of Sheffield, or King’s College London. The funders had no role in study design, data collection and analysis, decision to publish, or preparation of the manuscript. There was no additional external funding received for this study.”

3. For studies involving third-party data, we encourage authors to share any data specific to their analyses that they can legally distribute. PLOS recognizes, however, that authors may be using third-party data they do not have the rights to share. When third-party data cannot be publicly shared, authors must provide all information necessary for interested researchers to apply to gain access to the data. (https://journals.plos.org/plosone/s/data-availability#loc-acceptable-data-access-restrictions)

a) A description of the data set and the third-party source

b) If applicable, verification of permission to use the data set

c) Confirmation of whether the authors received any special privileges in accessing the data that other researchers would not have

d) All necessary contact information others would need to apply to gain access to the data.

Author response: Thank you for raising this. We have conducted a secondary analysis of data that is owned by a third party (the E-Risk Study) and thus we do not have permission to share it directly with other researchers. We have now provided a data availability statement within our revised manuscript with details of how other researchers can gain access to this data:

Data Availability Statement 

Due to the potentially identifying nature of these data, they are not publicly available. This and other E-Risk data can be accessed for free by researchers through a managed access process requiring an E-Risk Study sponsor. Full information on how to apply for access is available here: https://eriskstudy.com/data-access/ and queries should be emailed to Professor Helen Fisher at this address: eriskstudy@kcl.ac.uk

Author response: As mentioned above, due to the potentially identifying nature of these data, they are not publicly available. This and other E-Risk data can be accessed for free by researchers through a managed access process requiring an E-Risk Study sponsor. Full information on how to apply for access is available here: https://eriskstudy.com/data-access/ and queries should be emailed to Professor Helen Fisher at this address: eriskstudy@kcl.ac.uk We have now added this data availability statement to our revised manuscript.

Author response: We have now reviewed our reference list and ensured it is complete and accurate. It does not involve any retracted papers.

Additional Editor Comments (if provided):

This manuscript can be accepted for publication after minor edits:

1. more detail regarding how the data is preprosessed for the four machine learning algorithms should be presented in 2.4. 

2. suggestions by the reviewers,

In the meantime, the authors should also make sure the dataset used for this research can satisfy the requirement of Plos One as stated at: https://journals.plos.org/plosone/s/data-availability

Author response: We already have a paragraph about the pre-processing of the data in Section 2.4 as follows: “To align the audio segments to the speakers, Audacity was used. We used different tags to assign the segments of the interviewers and the mothers to general talk about both twins (e.g., the level of support during pregnancy), and specific talk about the elder and younger twins (e.g., feeling about her elder twin). In total, we had 38 distinct tags (19 for interviewers and 19 for mothers). Using the tags for the elder and younger twins, we divided the 52 recordings into 104 samples.”

In addition, we have now added the following sentences “Inaudible segments of the speech data were ignored. Due to having strong environmental background noises, we could not apply any noise reduction technique (causing loss of acoustic information).”

We have also addressed the other suggestions made by the reviewers (please see below) and added a data availability statement (please see above).

Reviewers' comments:

Reviewer's Responses to Questions

Comments to the Author

1. Is the manuscript technically sound, and do the data support the conclusions?

Reviewer #1: Yes

Reviewer #2: Partly

2. Has the statistical analysis been performed appropriately and rigorously?

Reviewer #1: Yes

Reviewer #2: Yes

3. Have the authors made all data underlying the findings in their manuscript fully available?

Reviewer #1: Yes

Reviewer #2: No

4. Is the manuscript presented in an intelligible fashion and written in standard English?

Reviewer #1: Yes

Reviewer #2: Yes

5. Review Comments to the Author

Reviewer #1: This is a fine paper. The methodology is worked out in a good way. They do not overinterpret their findings. The conclusions follow from the data and analysis.

The sample is somewhat small and the data are a bit old, but these aspects are indicated in the discussion. Results are promising indeed.

Author response: We thank the reviewer for their positive evaluation of our manuscript.

Reviewer #2: more literature review and recent papers can be refered . Technicall the Automatic detection of expressed emotion from Five Minute Speech is good . The result based insights can be included in the conclusion. In the experiment, the manuscript compares the results against Existing work.

It needs to justify why specifically these proposed work is compared

Author response: We have added the following paragraphs to Page 2 which include a wider review of the literature and more recent papers:

“There have been an increasing number of recent studies focusing on general emotion detection from speech. For instance, [14] applied a support vector machine classifier to detect verbal and nonverbal (e.g. laughing, crying) segments of speech and using deep residual networks, they extracted emotional and acoustic features from the segments. The feature embedding of the entire dialogue finally passed to an attentive long short-term memory (LSTM)-based classifier to detect emotions. Working on a Chinese dataset [15], the authors showed that the features extracted from the nonverbal segments could improve the performance of the classifier. [16] applied data augmentation (adding noise) to the RAVDESS dataset to improve the accuracy of a Convolutional Neural Network-based classifier detecting eight emotions (sadness, happiness, disgust, etc.). However, applying the same technique on a small local dataset (25 patients with stroke, dementia, epilepsy, etc.) did not yield improvement in the performance of their classifier. [17] fine-tuned two speech self-supervised automatic speech recognition models (Wav2vec 2.0 [18] and HuBERT [19]) on the IEMOCAP [20] dataset to detect emotions. The large HuBERT model outperformed the other models both in speaker-dependent and speaker-independent settings.These advanced state-of-the-art techniques, however, are not applicable in medical domains (including this study) where the number of data samples is limited, and the trained models could be easily overfitted. ”

Emotion is a continuous activity that psychologists categorise as discrete values such as anger, happiness, neutrality, etc. [23, 24]. Therefore SER could be interpreted as a regression problem or a classification model. As a regression model, the aim is to predict the emotion primitives such as arousal, valence, and dominance, while in a classification, they predict directly the discrete values [24]. Databases dedicated to SER depend on how the emotions are generated, which are generally acted or simulated, evoked or elicited, and emotions [23, 24]. To train the models different features could be extracted from the audio/video recordings including acoustic (prosody, spectral, voice quality features like jitters, shimmer, etc.) or non-acoustic (linguistic, disclosures, face and gestures). Most of the works are based on conventional classifiers (like SVM, and KNN), while there are some recent works on DNN-based models (e.g. CNN and LSTM) and transformer-based models. The conventional classifiers, which are dependent on feature engineering (normally easy to interpret), work faster especially for a small amount of data, while the DNN-based models do not rely on feature engineering (hard to interpret the features), requiring a large amount of data and computational resources [25]. They often are much slower than the conventional models. Lack of agreement on the definition of emotions, co-occurrence of additive noise in emotion, and differentiating between elicited, enacted, and natural emotion are challenges of SER. Other challenges are deciding on which feature selection techniques to apply, use or not use a complex DNN-based model, lack of enough training data, and data imbalance are other main challenges for SER [23, 24].

6. PLOS authors have the option to publish the peer review history of their article (what does this mean?). If published, this will include your full peer review and any attached files.

Do you want your identity to be public for this peer review? For information about this choice, including consent withdrawal, please see our Privacy Policy.

Reviewer #1: No

Reviewer #2: No

---

## [Editor Report · Decision Letter 1]

28 Feb 2024

Automatic detection of expressed emotion from Five-Minute Speech Samples: Challenges and opportunities

PONE-D-23-09462R1

Dear Dr. Mirheidari,

We’re pleased to inform you that your manuscript has been judged scientifically suitable for publication and will be formally accepted for publication once it meets all outstanding technical requirements.

Kind regards,

Jin Liu

Academic Editor

PLOS ONE

Additional Editor Comments (optional):

After carring out the reviewers suggestions, this manuscript can be accepted for publication now.
---

## [Editor Report · Acceptance letter]

7 Mar 2024

PONE-D-23-09462R1 

PLOS ONE

Dear Dr. Mirheidari, 

I'm pleased to inform you that your manuscript has been deemed suitable for publication in PLOS ONE. Congratulations! Your manuscript is now being handed over to our production team.

Kind regards, 

on behalf of

Professor Jin Liu 

Academic Editor

PLOS ONE